

# Translational effects of robot-mediated therapy in subacute stroke patients: an experimental evaluation of upper limb motor recovery

Eduardo Palermo[1], Darren Richard Hayes[1,2], Emanuele Francesco Russo[3], Rocco Salvatore Calabrò[4], Alessandra Pacilli[1] and Serena Filoni[3]

[1] Department of Mechanical and Aerospace Engineering, Sapienza University of Rome, Rome, Italy
[2] Seidenberg School of Computer Science and Information Systems, Pace University, New York, NY, USA
[3] Fondazione Centri di Riabilitazione Padre Pio Onlus, San Giovanni Rotondo, Italy
[4] IRCCS Centro Neurolesi 'Bonino-Pulejo', Messina, Italy

Corresponding author
Eduardo Palermo,
eduardo.palermo@uniroma1.it

## ABSTRACT

Robot-mediated therapies enhance the recovery of post-stroke patients with motor deficits. Repetitive and repeatable exercises are essential for rehabilitation following brain damage or other disorders that impact the central nervous system, as plasticity permits to reorganize its neural structure, fostering motor relearning. Despite the fact that so many studies claim the validity of robot-mediated therapy in post-stroke patient rehabilitation, it is still difficult to assess to what extent its adoption improves the efficacy of traditional therapy in daily life, and also because most of the studies involved planar robots. In this paper, we report the effects of a 20-session-rehabilitation project involving the Armeo Power robot, an assistive exoskeleton to perform 3D upper limb movements, in addition to conventional rehabilitation therapy, on 10 subacute stroke survivors. Patients were evaluated through clinical scales and a kinematic assessment of the upper limbs, both pre- and post-treatment. A set of indices based on the patients' 3D kinematic data, gathered from an optoelectronic system, was calculated. Statistical analysis showed a remarkable difference in most parameters between pre- and post-treatment. Significant correlations between the kinematic parameters and clinical scales were found. Our findings suggest that 3D robot-mediated rehabilitation, in addition to conventional therapy, could represent an effective means for the recovery of upper limb disability. Kinematic assessment may represent a valid tool for objectively evaluating the efficacy of the rehabilitation treatment.

# INTRODUCTION

Stroke, both ischemic and hemorrhagic, affects about 10 million people every year worldwide (*Vos et al., 2015*), representing the second most frequent cause of death,

after coronary artery disease and is the leading cause of disability in the elderly (*Wang et al., 2016*). Many stroke survivors (about 42 million in 2015) (*Vos et al., 2016*) sustain neurological damage, which is often permanent. Among other impairments, stroke can compromise use of the upper limbs, thereby negatively impacting common daily living activities (ADL) (*Papaleo et al., 2015*).

Although stroke patients are usually able to recover their ability to walk independently in a relatively short time, thanks to advanced rehabilitation therapies, a complete recovery of upper limb function is not as common (*Cho & Song, 2015*). Hence, effective therapies must be repetitive, target-oriented, and intense, in order to stimulate the neural plasticity processes, and are fundamental to the recovery of motor functionality (*Colombo et al., 2013*). Traditional therapy, manually administered by rehabilitation operators, rarely meets all of these criteria. The introduction of assisted robot therapy has improved the efficacy of upper limb rehabilitation and significantly improved the living conditions of patients (*Rahman et al., 2016*).

In recent years, technological advances, and increasing interest in robotic rehabilitation, have led to the development of high performance machines that can provide support to the rehabilitation operator; in some cases, they can even perform a perfectly complementary job (*Masiero et al., 2007*). Since the 1990s, these devices have become more pervasive. The first models allowed the operator to utilize pre-set tasks, and were activated '*as needed*' (*Squeri, Basteris & Sanguineti, 2011*), (*Reinkensmeyer, 2003*), thereby allowing the rehabilitation operator to follow multiple rehabilitation treatments simultaneously (*Wisneski & Johnson, 2007*). More recently, robots have integrated rehabilitation strategies that adapt to patient feedback. For example, robots can react to forces applied by the patient during rehabilitation (*Kahn et al., 2006*).

An essential feature of robotic devices is the ability to perform repetitive movements over a long period of time. The repetition and intensity of exercises are crucial in rehabilitative therapies for patients affected by stroke or other neurological pathologies. Research has shown that neural plasticity is preserved after a brain injury, thereby allowing for new connections to form between the neurons while their gradual reorganization can restore movement and functionality to the affected limb (*Turner et al., 2013*). Thanks to the virtual environments where exercises are performed, in the form of games with specific goals, the patient is more immersed compared to traditional therapies, which constitutes a further benefit.

Noteworthy, examples of upper limb rehabilitation robots currently available on the market or in research laboratories include the MIT-Manus for the end-effector typology (*Hogan et al., 1992, 1993*), and the Armeo®Power exoskeleton (Hocoma, Inc., Volketswil, Switzerland; https://www.hocoma.com/us/solutions/armeo-power/), which is derived from the research prototype Armin (*Mihelj, Nef & Riener, 2006*; *Nef et al., 2006*). The latter has been involved in studies that introduced a novel rehabilitation solution to foster neural plasticity, which showed promising results derived from transcranial magnetic stimulation (*Calabrò et al., 2016*).

Rehabilitation mediated by robots also provides quantitative results about improvements in task execution, thereby allowing researchers to quantitatively monitor
the recovery of limb functionality (*Panarese et al., 2016*). These performance indicators represent a fundamental method to evaluate the administration of specific rehabilitation protocols or the prescription of different exercises during rehabilitation. Performance data used to estimate the patient's motion capabilities can be obtained during specific exercises, via software installed on the device.

Considering this potential, robot-mediated therapy (RMT) became prominent in research activities that were focused on improving traditional rehabilitation paradigms (*Stein, 2004*). Although many studies to date have reported on the recovery of post-stroke patients treated through RMT, it is still difficult to assess the extent to which these results go beyond traditional therapies administered within a comparable timeframe (*Norouzi-Gheidari, Archambault & Fung, 2012*). In other words, despite the greater level and quality of both support and stimulation provided to patient, and the evaluating tools made available to clinicians, demonstrating higher effectiveness in recovery of RMT with respect to traditional therapy is still an open challenge.

One reason for this lack of evidence lies in the heterogeneity of RMT solutions and, consequently, in the wide variety of strategies that have been proposed to evaluate its effects. To analyse pre-post treatment effects, many studies have combined a kinematic evaluation of patients' motor performance compared to traditional evaluation techniques, in an effort to overcome the intrinsic challenges associated with replicating clinical scales (*Prange et al., 2006*). In fact, despite being designed to comprehensively evaluate different aspects of motor deficit resulting from a stroke, clinical scales are prone to uncommon sensitivity, ceiling effects, and subjectivity in their administration by the operator (*McCrea, Eng & Hodgson, 2002*; *Rohrer et al., 2002*; *Bosecker et al., 2009*). However, in most cases, this type of supplementary metric is calculated on the same gestures performed for the treatment. Consequently, in order to reveal the translational effects of rehabilitation, an evaluation of the motor performance regained by the patients should involve gestures that mirror daily activities, and are derived from the RMT scenario (*Van Kordelaar, Van Wegen & Kwakkel, 2014*).

Several studies have assessed the possibility of using a kinematic evaluation based on a simple daily-life inspired gesture, to objectively assess stroke-related motor impairment. *Van Kordelaar, Van Wegen & Kwakkel (2014)* proposed evaluating kinematic parameters that are based on the hand trajectory recorded by an electromagnetic motion tracking device, during a simple exercise based on reaching and moving objects on a table. A similar paradigm was proposed by *Rohrer et al. (2002)* that assessed motion smoothness changes during recovery in the aftermath of a stroke by leveraging a MIT-Manus. However, these movements are planar and the gravity load effect is supported by the robot or by the table. In contrast, a 3-DoF protocol would facilitate an evaluation of the final effect of recovery, where the force exerted for a vertical elevation of the hand plays an important role. *Caimmi et al. (2007)* proposed a 3-DoF protocol where the subject was tasked with reaching towards a target placed in front of him at shoulder level and starting from a lower position. Kinematic indices, based on motion capturing, demonstrated improvements for stroke survivors thanks to constraint-induced movement therapy (*Caimmi et al., 2007*).

In this paper, we adopted a protocol similar to the one introduced in *Caimmi et al. (2007)*, for evaluating the translational effects of an RMT-based rehabilitation project and administrated to 10 stroke survivors, using a rehabilitation exoskeleton: the ARMEO Power device. In particular, we primarily sought to investigate whether kinematic indices, based on motion capturing a 3D daily-life inspired gesture, improved after the administration of an RMT protocol, which involved an exoskeleton for 3D upper limb rehabilitation. As a secondary goal, we evaluated how these indices are in agreement with patient assessments that have been assessed using the most widely adopted clinical scales for post-stroke motor impairment.

## METHODS

### Patients' description

A total of 10 subjects (eight males and two females, mean age 60.1 ± 18.3 years) affected by stroke in the sub-acute phase (4.0 ± 1.5 months after the event; five with left and five with right hemiparesis) were enrolled in the study.

Inclusion criteria were:

- unilateral paresis from a single supratentorial stroke occurring at least six months prior;
- sufficient cognition to follow simple instructions and understand the purpose of the study (Mini-Mental State Examination score > 18 points) (*Masiero et al., 2007*);
- ability to perform the task proposed (pointing at a target, with the unaffected and with affected limb);
- ability to remain in a sitting posture.

Exclusion criteria were:

- participation in other studies or rehabilitation programmes;
- bilateral impairment;
- severe spasticity (Modified Ashworth Scale score $\geq$ 3);
- severe sensory deficits in the paretic upper limb;
- other neurological, neuromuscular or orthopedic (shoulder sub-luxation or pain in the upper limb) disorders, or visual deficit;
- refusal or inability to provide informed consent;
- other concurrent severe medical problems.

Table 1 reports the primary clinical data of the patients included in the study. Patients were clinically evaluated using the four most adopted rating scales in stroke: the motor sub-section of the Functional Independence Measure (FIM) (*Liao et al., 2012*), Barthel Index (BI) (*Parker, Wade & Hewer, 1986*), Frenchay Arm Test (FAT) (*Heller et al., 1987*), and Fugl-Meyer Assessment (FMA, Motor function sections, maximum score 66) (*Sullivan et al., 2011*).

**Table 1 Clinical data of the patients' population.**

| Patient | Gender | Age | Affected side | Months after event |
|---|---|---|---|---|
| 1 | M | 66 | Left | 5 |
| 2 | F | 56 | Right | 2 |
| 3 | M | 40 | Left | 5 |
| 4 | M | 74 | Left | 5 |
| 5 | M | 73 | Right | 4 |
| 6 | M | 54 | Right | 4 |
| 7 | M | 65 | Left | 2 |
| 8 | M | 21 | Right | 2 |
| 9 | M | 69 | Right | 6 |
| 10 | F | 83 | Left | 5 |

## Treatment protocol and device

This study was performed in accordance with the Declaration of Helsinki and was approved by the ethics committees of IRCCS Centro neurolesi Bonino Pulejo (study registration number 43/2013). Informed consent was obtained from all subjects enrolled in this study.

Patients underwent a rehabilitation programme of 20 sessions, each lasting 50 min, five sessions per week, using the Armeo®Power exoskeleton, in addition to a session of conventional rehabilitation therapies conducted with the same duration.

The Armeo®Power (Fig. 1) is a motorized orthosis for the upper limb with six degrees of freedom (DoFs): three DoFs for the shoulder, one for the elbow flexion, one for the forearm supination, and one for the wrist flexion. Each joint is powered by a motor and equipped with two angle sensors.

The device can support the patient's arm weight, thereby providing a feeling of fluctuation, and assists it in a large 3D workspace during execution of the exercises. The presence of a suspension system allows the facilitator to set and adjust the sensitivity of the robot depending on the characteristics of each patient. Arm and forearm lengths are both adjustable, so that the device can be adapted for use by a large selection of patients.

The interface used for the execution of exercises, which appear in the form of games, is designed to simulate arm gestures and provide a simple virtual environment. Increasing levels of difficulty can be selected, which in turn determines the speed of the movements, their direction and the work area, depending on the degree of motility of the subject undergoing rehabilitation.

Each robotic session, which lasted 50 min, consisted of: 10 min of passive mobilization for familiarization and to decrease the patient's spasticity, if present; and 40 min of task-oriented exercises that were calibrated according to the patient's abilities and with increasing difficulty over the course of the training period.

## Experimental setup for kinematic analysis

To evaluate the effects of the prescribed treatment, patients underwent a 3D kinematic analysis, both pre- and post-treatment. Patient movements were recorded during a pointing task (Fig. 2), using an Optoelectronic System, the BTS SMART-DX 300

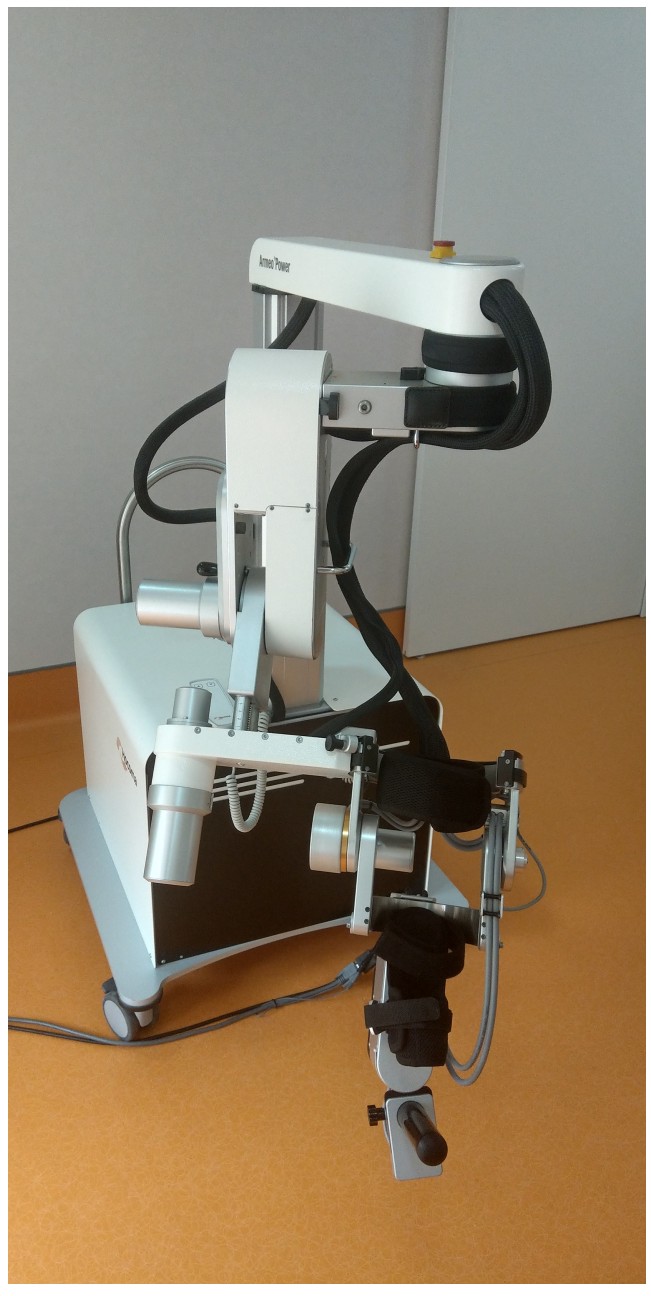

**Figure 1 The Hocoma Armeo®Power.** A six degrees of freedom (DoFs) exoskeleton: three DoFs for the shoulder, one for the elbow flexion, one for the forearm supination, and one for the wrist flexion. Each joint is powered by a motor and equipped with two angle sensors (Photo: E F Russo).

(BTS Bioengineering, Brooklyn, NY, USA; http://www.btsbioengineering.com), which consists of six infrared CCD cameras with a resolution of 650 × 480 pixels, and an acquisition rate of 120 Hz.

In this study, a subset of the kinematic model proposed by *Rab, Petuskey & Bagley (2002)* has been adopted, to ensure the optimal execution of the exercise. In particular, the head, neck, and pelvis segments have been removed. Reflective markers were placed on

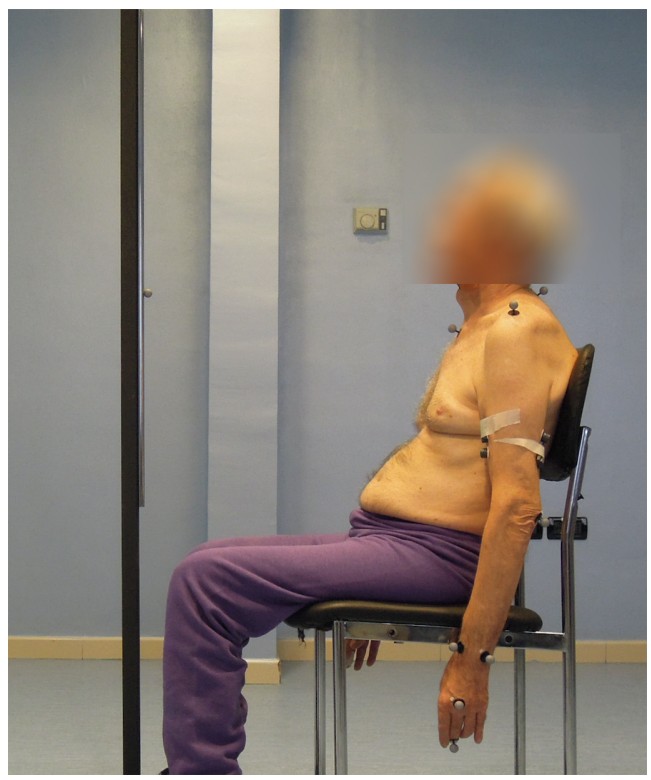

**Figure 2 Subject performing the reaching task.** Reflective markers are placed on the body according to the kinematic model adopted. A reflective marker is also placed on the target, installed on a rod in front of the subject, at the height of the shoulders. The rod was moved at every trial to align the target with the shoulder performing the reaching task (Photo: E F Russo).

the patient's body, referenced in the model (Fig. 3). The wrist joint is modelled as a universal (saddle) joint with two DoFs, where wrist movement occurs in flexion-extension and radio-ulnar deviation; the elbow like a hinge joint with two DoFs; the shoulder as a spherical joint with three DoFs.

The pointing task designed for 3D kinematics acquisition required reaching a target, placed on the subject's sagittal plane, at shoulder height, and at a distance from the body equal to the patient's arm length (measured from the acromion marker to the midpoint between the radius and ulna markers). The patient was sitting on a chair with his hands stretched along his hips and his back resting but not locked in that position, thereby allowing compensatory movements, which were also measured (Fig. 2).

Each of the two kinematic evaluations (pre- and post-treatment) were recorded in a session in which the patient was invited to reach and point at the target from a neutral position, without straining, first with the healthy limb and then with the paretic one. Repeating the task six times took about 10 min.

## Data processing and statistical analysis

Three-dimensional marker trajectories were recorded using frame-by-frame acquisition software (SMART Capture—BTS, Milan, Italy) and labelled using frame-by-frame

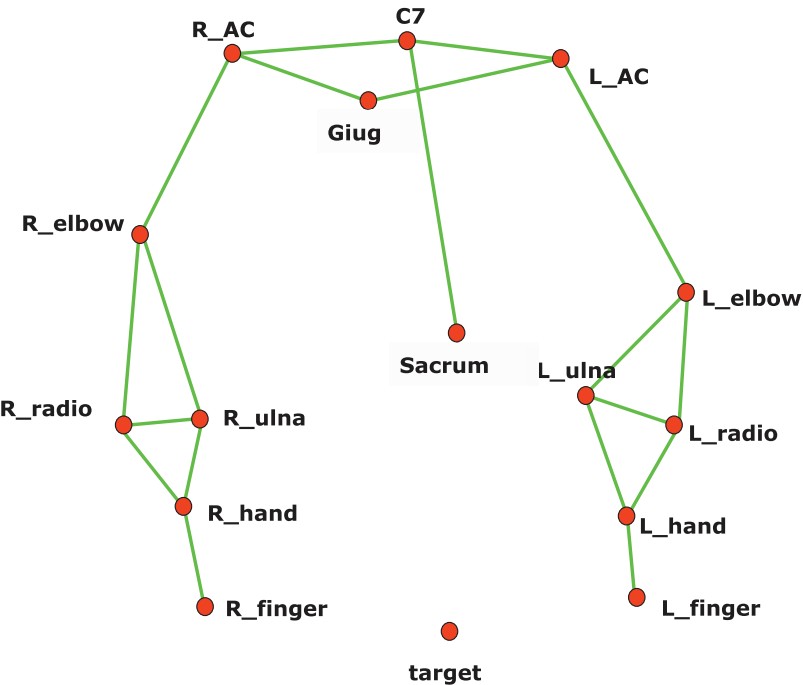

**Figure 3 Kinematic model for reflective marker placement adopted in this study.** A total of 12 markers (14 mm diameter) are placed over prominent bony landmarks of the upper extremity, easily identifiable, and reproducible, where subcutaneous tissue is thin, minimizing soft tissue artifact due to marker movement with respect to bone.

tracking software (SMART Tracker—BTS, Milan, Italy). The captured data were transferred to MATLAB software (The Mathwors Inc., Natick, MA, USA), were interpolated and filtered with a 6 Hz second-order Butterworth filter in both forwards and reverse directions, resulting in a zero-phase distortion and fourth order filtering.

The velocity of the hand marker was computed using numerical differentiation. Movement onset was defined at the time when the velocity of the hand marker exceeded 5% of the maximum velocity in the pointing phase. Movement offset was detected when the velocity of the hand was below the threshold previously described (*Alt Murphy, Willén & Sunnerhagen, 2012*).

Kinematic data of the session were processed to calculate the following indices:

- Movement time (MT), as the total execution time of the task (between onset and offset), measured in seconds (*Cho & Song, 2015*; *Alt Murphy, 2013*; *Frisoli et al., 2012*; *Burgar et al., 2011*);
- Peak velocity (PV), as the maximum value of the speed profile curve of the hand marker, measured in meters per second (*Alt Murphy, 2013*; *Subramanian et al., 2010*; *Coscia et al., 2014*; *Bartolo et al., 2014*; *Rigoldi et al., 2012*; *Menegoni et al., 2009*);
- Time to PV (TtPV), as the percentage of time from the beginning of the movement to the peak speed (*Alt Murphy, 2013*; *Rigoldi et al., 2012*);
- Normalized Jerk (NJ), as a non-dimensional quantity which corresponds to the square root of the jerk (third derivative of the position of the hand marker with respect to time),

mediated over the entire duration of the movement, and normalized with respect to MT and to the total displacement of the onset and offsets (L) (*Coscia et al., 2014*; *Bartolo et al., 2014*; *Bland & Altman, 1994*);

- Trunk Displacement (TD), measured in meters to identify compensation movements, calculated as the difference between the maximum displacement of the trunk marker and its initial position in space, normalized with respect to distance C7-sacrum, expressed as a percentage (*Alt Murphy, 2013*; *Subramanian et al., 2010*; *Coscia et al., 2014*);

- Hand Path Ratio (HPR) is the ratio of the distance travelled by the hand between the movement onset and offset and the straight-line distance between the starting and destination targets, expressed as a percentage (*Subramanian et al., 2010*; *Rigoldi et al., 2012*; *Menegoni et al., 2009*; *Colombo et al., 2005*).

Movement time, PV, and TtPV indices are related to the time required for pointing at the target and the speed at which the task is performed.

Normalized Jerk quantifies the fluidity of motion: higher values correspond to lower smoothness, reflecting poor fluidity in motion, or absence of fine tuning of muscular control, whereas a fluid movement will be expressed by a lower value. Although other indices of smoothness have been proven valuable during the last few years (*Balasubramanian et al., 2015*), today NJ is the most widely adopted index for smoothness.

Trunk Displacement provides information about the compensation strategies implemented by the patient during execution of the task.

Hand Path Ratio, instead is considered an index of motion accuracy in point-to-point movements (*Do Tran, Dario & Mazzoleni, 2018*).

Statistical analyses were performed with SPSS software (Statistical Packages for Social Sciences, version 24.0; SPSS Inc., Chicago, IL, USA). Considering the non-normal distribution of the indices and the small size of the sample, non-parametric tests with a 95% confidence interval ($\alpha = 0.05$) were applied. In particular, the Wilcoxon signed-rank test two-tailed was chosen to verify whether there were differences between pre- and post-treatment for each parameter. The Spearman correlation test was performed to highlight any correlation between kinematic parameters and the main clinical scales used.

## RESULTS

An example of reaching trajectories, obtained during the evaluation tasks, is depicted in Fig. 4. Figure 5 reports mean and standard deviation values of the NJ calculated on hand trajectories during each task repetition. Significant differences between pre- and post-treatment kinematic indices were found for MT ($Z = -2.701$, $p = 0.007$), NJ ($Z = -2.701$, $p = 0.007$), TD ($Z = -2.701$, $p = 0.007$), and HPR ($Z = -2.701$, $p = 0.007$). The average values of all these parameters were lower after the treatment than before, as reported in Fig. 6. No significant difference was found for PV, and TtPV between the pre- and post-treatment evaluations. As displayed in Figs. 5 and 6, the values of the indices, derived from the affected arm, are reported along with values obtained from the unaffected arm, to visually compare the difference in the indices and illustrate improvement.

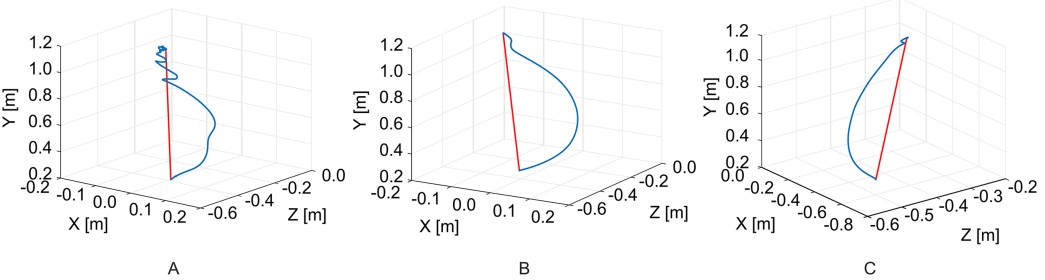

**Figure 4 Example of hand trajectories.** Hand trajectories (in blue) during the reaching task, with respect to the shortest path (in red). (A) Reaching trajectory of the paretic arm before the treatment. (B) Reaching trajectory of the paretic arm after the treatment. (C) Trajectory of the unaffected arm.

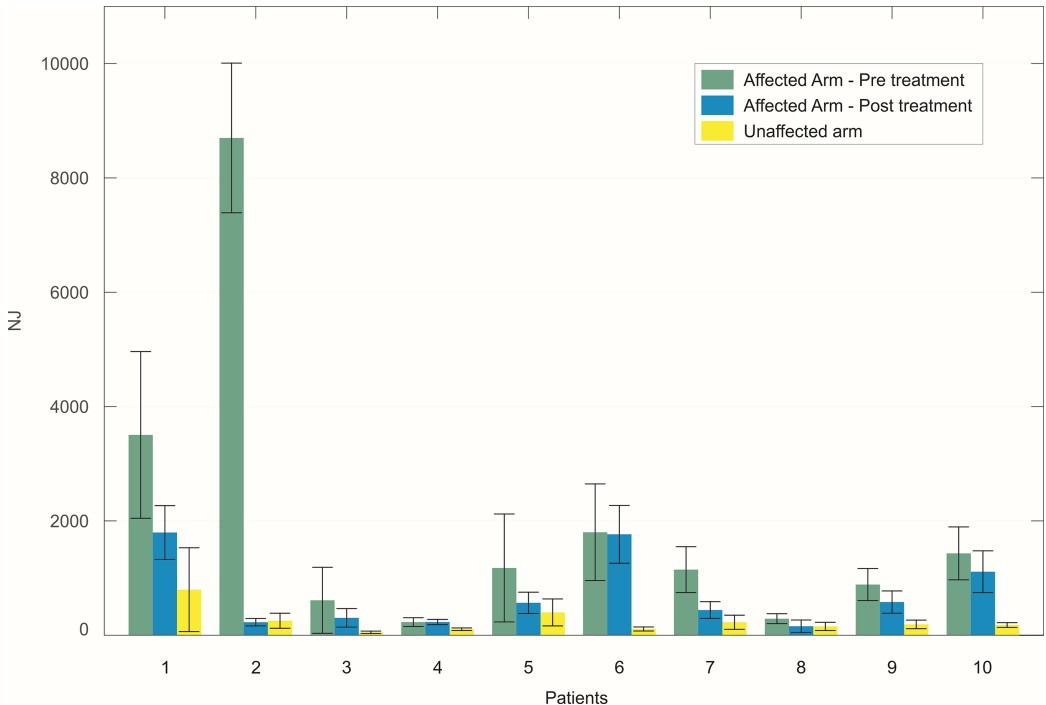

**Figure 5 Mean value of the NJ of hand trajectories for the ten patients across the different trials.** The bars represent the standard deviation (±). For each patient, values obtained with the affected arm before the treatment (green) are compared to those obtained with the same arm after the treatment (blue). Values obtained with the unaffected arm are also reported for visual comparison (yellow).

All of the administered clinical assessment scales resulted in pre- vs. post-treatment significant decrease: FIM ($Z = -2.803$, $p = 0.005$), BI ($Z = -2.809$, $p = 0.005$), FAT ($Z = -2.831$, $p = 0.005$), FMA ($Z = -2.807$, $p = 0.005$), as reported in Table 2.

Table 3 reports the results of the Spearman correlation test, across all kinematic parameters and all administered clinical assessment scales. A strong tendentially significant correlation was found between FAT and HPR. A moderate, yet not significant, correlation ($0.40 < |rs| < 0.59$), was found between BI and MT, BI and TD, FAT and TtPV, and FMA and HPR.

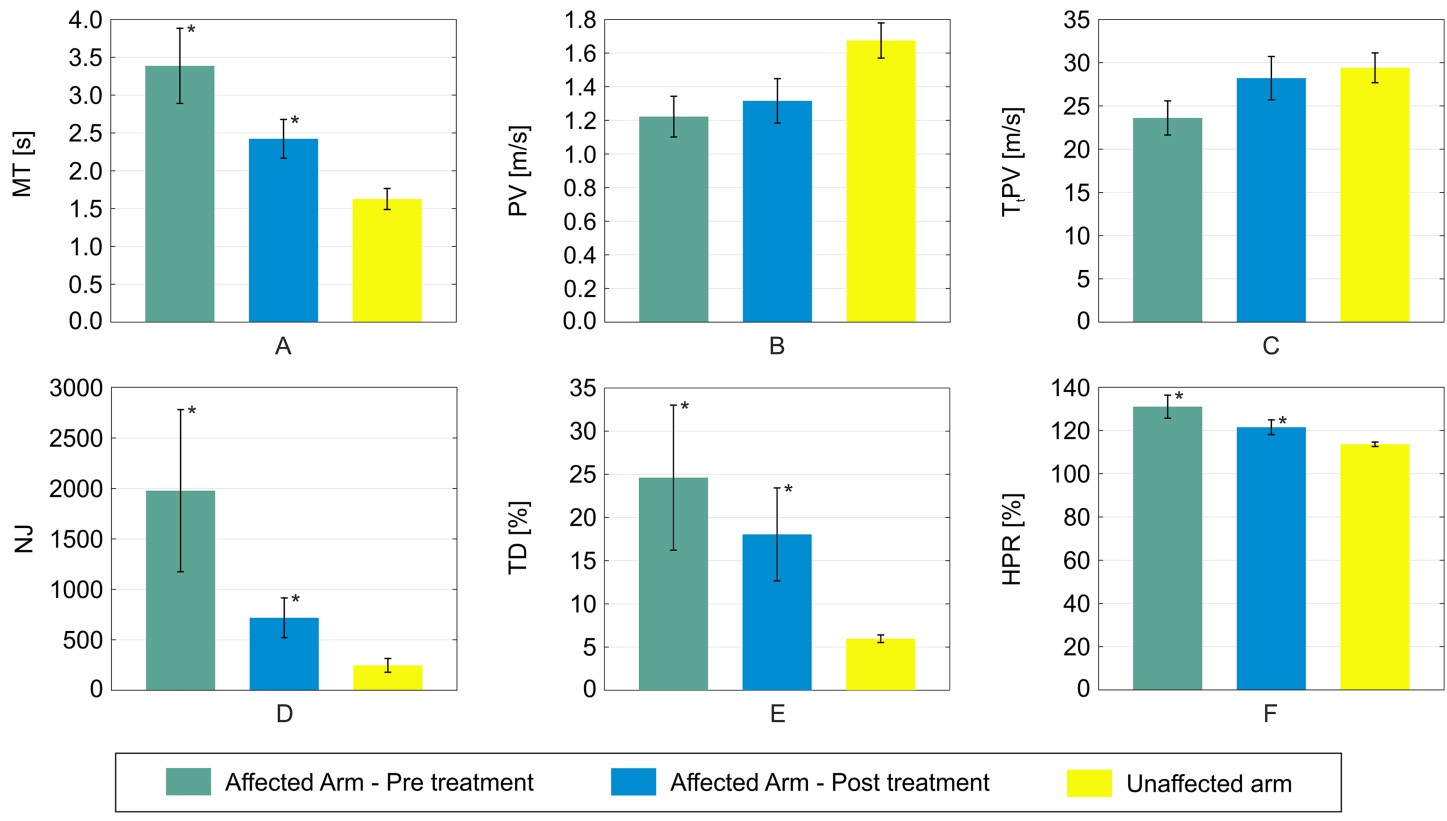

**Figure 6 Mean values of the six kinematic indices calculated across all the patients.** (A) MT; (B) PV; (C) TtPV; (D) NJ; (E) TD; (F) HPR. Error bars represent the standard deviation (±). For each index, mean values obtained with the affected arm before the treatment are depicted in green. Values obtained with the same arm after the treatment are reported in blue. Statistical significance between the two conditions are starred. For visual comparison, values obtained with the non-affected arm are also reported in yellow.

**Table 2 Spearman correlation coefficients and significance level (in brackets) between the four clinical scales score and kinematic parameters, evaluated post-treatment.**

| Scale | Pre-treatment | Post-treatment | $p$ value |
|---|---|---|---|
| FIM | 78.5 ± 19.1 | 98.7 ± 13.6 | 0.005 |
| BI | 52.5 ± 21.1 | 75.5 ± 14.0 | 0.005 |
| FAT | 1.5 ± 1.4 | 4.2 ± 1.1 | 0.005 |
| FMA | 32.6 ± 13.9 | 45 ± 10.7 | 0.005 |

**Table 3 Pre-treatment and post-treatment values (mean ± standard deviation) of clinical scales. Superscript T marks tendentially significant values.**

| | Index | | | | | |
|---|---|---|---|---|---|---|
| Scale | MT | PV | TtPV | NJ | TD | HPR |
| FM | −0.164 (0.650) | 0.024 (0.947) | 0.359 (0.309) | −0.207 (0.567) | −0.140 (0.699) | −0.049 (0.894) |
| BI | −0.470 (0.171) | 0.384 (0.273) | −0.049 (0.894) | −0.396 (0.257) | −0.511 (0.131) | 0.024 (0.947) |
| FAT | −0.192 (0.595) | −0.096 (0.792) | 0.528 (0.117) | −0.329 (0.353) | 0.364 (0.301) | −0.624[T] 0.054 |
| FMA | −0.036 (0.920) | −0.120 (0.973) | 0.164 (0.650) | 0 1 | 0.152 (0.674) | −0.426 (0.220) |

## DISCUSSION

In this study, we analysed the effects of RMT conducted with an exoskeleton that supported the 3D movement of the upper limbs, involved 10 stroke survivors, using a pre- vs. post-treatment 3D kinematic analysis of a specific upper limb gesture, which mirrored a daily living activity. Their residual motion capabilities were evaluated by means of a set of kinematic parameters that were measured during execution of a reaching task with both the paretic and the unaffected arm, other than by using the four most adopted clinical scales.

Our findings demonstrate the benefits of a rehabilitation programme focused on the range of motion capabilities of post-stroke patients. Indeed, these patients demonstrated an improvement across all administered clinical scales, and these results are in agreement with the kinematic analysis conducted. The trajectories of reaching tasks performed after treatment were smoother and more accurate. Four out of the six kinematic indices computed on the reaching trajectories travelled after the treatment of the paretic arm were different to those obtained before the treatment. In particular, indices obtained with the paretic arm, after the treatment, showed movement more comparable to the unaffected arm.

The significant decrease in MT indicates regained mobility with gesture performance. A reduced time to complete the task implies a more effective combination of motion smoothness and accuracy. Regardless of the actual distribution of improvement from these two aspects, the overall ability to complete the task in a reduced timeframe indicates an increase in patient independence in daily life, which is a key concern in rehabilitation. *Frisoli et al. (2012)* demonstrated a significant correlation with the total time for the reaching movement with the clinical evaluation of motor impairment in both ischemic and hemorrhagic stroke patients. This index showed a significant decrease after a rehabilitation programme, towards the value observed in healthy control group.

Normalized Jerk is generally understood as an index of motion smoothness, where higher levels of this parameter are typical of less smoothly controlled gestures (*Hogan & Sternad, 2009*). All of the patients showed a noticeable decrease in the NJ average value in reaching tasks performed with the paretic arm. Values of NJ obtained after the RMT programme, are closer to those performed with the unaffected arm.

Conversely, HPR represents the subject's ability to perform a reaching trajectory within the shortest possible distance between the start and target points. A line connecting the two points does not exactly represent the path chosen by unimpaired subjects, as shown in Fig. 3. However, the difference between the actual hand trajectory and the line is an important parameter in evaluating accuracy in reaching tasks (*Burgar et al., 2011*). Stroke survivors who participated in our study exhibited a significant decrease in this index in post-treatment trials compared to pre-treatment ones, with an average value more comparable with that of the unaffected arm.

Another significant improvement was observed in the TD index for our sample study. This result can be interpreted as a secondary effect of the restoration of motor activation paths from the motor cortex to muscles. The increased capability of the subject

to fire the necessary motor units required less compensatory trunk muscle activity to complete the task (*Murphy, Willén & Sunnerhagen, 2010*). *Murphy, Willén & Sunnerhagen (2010)* demonstrated that TD is significantly higher in post-stoke patients than in healthy subjects, and a noteable increase in this index can also be observed between patients with moderate stroke with respect to those with a mild stroke.

Interestingly, no significant effect was observed in PV and TtPV. Although these are generally considered indices of motor capability in point-to-point tasks (*Alt Murphy, 2013*), the patients involved in this study did not exhibit any significant variation in these two indices. The restored motor control, highlighted by other observed markers, both clinical and kinematic, was not reflected in the velocity profile of the hand during specific pointing tasks. Thus, our preliminary findings suggests that one should not simply rely on these two indices as effective measurements for the effectiveness of a rehabilitation programme.

Although the results were obtained from a small patient sample, the findings of the present study are particularly important to current discussions about RMT. Moreover, to date, several studies have assessed improvements in the motion capabilities of stroke survivors after RMT treatments, for a larger cohort of subjects (*Lo et al., 2009*). However, improvement has mainly been evaluated by means of clinical rating scales or kinematic indices computed on gesture trajectories performed during rehabilitation treatment. Thus, it is generally accepted that training stroke survivors to perform specific upper arm trajectories, in a controlled and assisted manner that is facilitated by a robotic device, leads to improvement in performing a specific task. However, a key issue with motor rehabilitation is the translational effect of therapy, that is the potential to improve gestures typically associated with daily life, distinct from those performed in a rehabilitation programme (*Kwakkel et al., 1997*). This problem is of great importance in analysing the RMT effect throughout the entire rehabilitation process. Current studies only merely use the robotic device as a theraputic instrument, while the kinematic evaluation was used to evaluate a simple gesture, which was highly representative of a daily life scenario.

Despite the number of RMT studies conducted thus far, proving increased performance compared to a traditional rehabilitation programme, within the same timeframe, remains a challenge (*Norouzi-Gheidari, Archambault & Fung, 2012*). One reason is the lack of a standardized evaluation protocol for measuring the impact, apart from the use of clinical scales. Although highly comprehensive and well structured, clinical scales are not an objective tool, and are often comprised of different characteristics related to disability, ranging from motor capabilities to facial expressions or psychological treats. The protocol presented in this study has the potential to serve as a standard evaluation tool for more objectively quantifying upper limb motor smoothness and accuracy, derived from a rehabilitation programme, and ultimately inspiring comparative studies on the efficacy of RMT vs. traditional therapy. Several studies have examined the pointing movement in stroke patients (*Alt Murphy, 2013*; *Frisoli et al., 2012*; *Subramanian et al., 2010*; *Duret & Hutin, 2013*; *Cirstea & Levin, 2000*; *Nordin, Xie & Wünsche, 2014*); however, they have used different kinematic variables to analyse the movement, despite the common goal of being able to quantify speed, accuracy and fluidity of movement.

In this vein, a comparative analysis of patient behaviour in kinematic evaluation, in terms of clinical scales score, is of great importance.

The kinematic evaluation protocol that we adopted, instead, was introduced by *Caimmi et al. (2007)*, to evaluate the effects of constraint-induced therapy. In this study, we used this method to evaluate effects of RMT sessions, performed using a rehabilitation exoskeleton that induces 3D movements of the upper limb. Reporting these findings is valuable as the literature is lacking when it comes to these types of studies, especially with RMT solutions inducing planar movements, where the gravity effect is completely supported. The adoption of 3D robotics, assisting the subjects in compensating for gravity, is expected to enhance this capability.

To consolidate the preliminary findings of this study, and positively contribute to the current discussions about the impact of RMT, future studies should involve a larger patient sample, in parallel with a control group undergoing conventional therapy. If confirmed on a larger number of patients, the positive results reported herein will pave the way for the establishment of a standardized procedure for objectively evaluating motor recovery in conjunction with a robotic rehabilitation programme. This tool would have a tremendous potential in facilitating comparative studies about the effects of RMT compared to traditional physical therapy for rehabilitation.

Moreover, apart from the advantaged documented in this paper, as in other similar studies, it is not possible to isolate the effects of RMT per se. A physiological progressive improvement in the motor capabilities of stroke survivors, during the subacute phase, has already been demonstrated (*Van Kordelaar, Van Wegen & Kwakkel, 2014*). Thus, a comparative study with only two groups of stroke survivors would be required, where one group is treated with RMT, which would accurately quantify the benefits of RMT, although this would be questionable in terms of ethics. Ultimately, reporting the results of a specific therapy, using a standard protocol and a set of accepted indices, is valuable, as it permits a better interpretation of the actual outcomes of the therapy.

## CONCLUSIONS

In this study, we analysed the effects of robot-mediated therapy on 10 stroke survivors, through a pre- vs. post-treatment 3D kinematic analysis of a specific upper limb gesture, simulating ADL. Their residual motion capabilities were evaluated by means of a set of kinematic parameters measured during the execution of a reaching task with both a paretic and an unaffected arm.

Our results highlighted the efficacy of a rehabilitation programme that benefits the motion capabilities of patients. Patients exhibited improvements in all of the administered clinical scales, which was in agreement with the kinematic analysis conducted.

Although the analysis was obtained from a small sample of patients, the findings of our study have the potential to contribute to the current discussions about robot-mediated therapy. The protocol presented in this study, inspired by daily-life gestures (upper limb motor tasks), may represent a step forwards in establishing a standard evaluation procedure, for the objective quantification of upper limb motor recovery following RMT-based treatments.

### Funding

The authors received no funding for this work.

### Competing Interests

The authors declare that they have no competing interests.

### Author Contributions

- Eduardo Palermo conceived and designed the experiments, analyzed the data, contributed reagents/materials/analysis tools, prepared figures and/or tables, authored or reviewed drafts of the paper, approved the final draft.
- Darren Richard Hayes conceived and designed the experiments, analyzed the data, contributed reagents/materials/analysis tools, authored or reviewed drafts of the paper, approved the final draft.
- Emanuele Francesco Russo conceived and designed the experiments, performed the experiments, analyzed the data, contributed reagents/materials/analysis tools, prepared figures and/or tables, authored or reviewed drafts of the paper, approved the final draft.
- Rocco Salvatore Calabrò conceived and designed the experiments, contributed reagents/materials/analysis tools, authored or reviewed drafts of the paper, approved the final draft.
- Alessandra Pacilli conceived and designed the experiments, contributed reagents/materials/analysis tools, authored or reviewed drafts of the paper, approved the final draft.
- Serena Filoni conceived and designed the experiments, performed the experiments, contributed reagents/materials/analysis tools, authored or reviewed drafts of the paper, approved the final draft.

### Human Ethics

The following information was supplied relating to ethical approvals (i.e. approving body and any reference numbers):

This study was performed in accordance with the Declaration of Helsinki and was approved by the ethics committees of IRCCS Centro neurolesi Bonino Pulejo (study registration number 43/2013).

### Data Availability

The raw data are also provided in a Supplemental File.

### Supplemental Information

Supplemental information for this article can be found online at http://dx.doi.org/10.7717/peerj.5544#supplemental-information.

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
