# Peer review of "Translational effects of robot-mediated therapy in subacute stroke patients: an experimental evaluation of upper limb motor recovery"

_PeerJ, doi:10.7717/peerj.5544_

## Round 0.1 · original submission · Major Revisions

Dear Authors,

Enclosed are important suggestions by two peer reviewers which indicates major revision to your submitted manuscript to PeerJ.

Thank you

Reviewer 1 ·

Basic reporting

1. The article uses clear, unambiguous and technically correct text. It conforms to professional standards of courtesy and expression.
2. The article includes sufficient introduction and background to demonstrate how the work fits into the broader field of knowledge. I suggest the authors to enrich what they state in lines 51-55 with more references.
3. The structure of the article conforms to an acceptable format of ‘standard sections’.
4. Figures are relevant to the content of the article, of sufficient resolution, but I suggest the authors to increase the description of them. Moreover, a figure showing the markers location would help the reader. Otherwise, the authors could show markers locations by figure 2, but comprehensive description is needed.
5. All appropriate raw data has been made available in accordance with Data Sharing policy.

Experimental design

1. The research question is not well defined
2. Methods describe no sufficient information to be reproducible by another investigator.
• What kind of pre-processing did the authors use?
• How onset and offset of movement were defined?
3. Could the authors explain me how HPR parameter provide information about the compensation strategy?
4. Why the authors did not consider an accuracy index?

Validity of the findings

1. The study could be considered a meaningful replication. The parameters are widely accepted but the authors did not consider some parameters such as accuracy (please, refer to Balasubramanian et al (2012)). There is no novelty in the analysis proposed. I would spent some words about the motion smoothness measure. Normalized jerk is, probably the most common measure of smoothness. However, recent studies have suggested other type of parameters such as SPARC (see Balasubramanian et al. (2015)).

2. Data are robust, statistically sound, & controlled. However, I believe that showing the arm in the results, without having considered it in the statistical design, is confusing.

a. Figure 5. What are the error bar?
b. Table 3. Please explain what symbols mean.

Additional comments

MAJOR
- The submission do not clearly define the research question. It is not clear, if the authors contributes to increase the knowledge gap of robot mediated therapy efficacy or if they propose the kinematic evaluation instead of/in support of clinical scales. Which are the hypothesis of this study?
- Introduction: I suggest clearly declaring which is the primary outcome of the study and the secondary outcomes. If authors’ intention was to proof the efficacy of RMT, then the study lacks in the design of the experiment because of the absence of a control group undergoing only a conventional rehabilitation therapy. While if the intention was to propose the kinematic evaluation of motor performance, then I would spend more words about the advantages and disadvantages of clinical scales assessing motor performance and describing what is the state of the art about quantitative assessment of motor performance.
- Methods: Are movements comparable between subjects? Did the authors normalize with respect to arm length, time or something else?
- Because of the confusion, defining the original research question, Discussion section is difficult to read.
a. First, I suggest to briefly describe the scientific questions declared in the introduction.
b. Line 227. How can the authors assert that the affected arm is comparable with the unaffected arm? Maybe it would be sufficient to point out that this is just a qualitative impression and that obviously a further quantitative analysis is needed.
c. Line 285. The authors did not measure accuracy.
d. Lines 287-291. This paper would acquire more value if the authors had used a system more ambulatory affordable. I agree with the authors that clinical scales could lack in objectivity but I believe that the alternative should be low cost, easy-to-use, reliable, etc..

- Please find here some useful references:
o Reinkensmeyer, D. J. "How to retrain movement after neurologic injury: a computational rationale for incorporating robot (or therapist) assistance." In Engineering in Medicine and Biology Society, 2003. Proceedings of the 25th Annual International Conference of the IEEE, vol. 2, pp. 1479-1482. IEEE, 2003.
o Squeri, Valentina, Angelo Basteris, and Vittorio Sanguineti. "Adaptive regulation of assistance ‘as needed’ in robot-assisted motor skill learning and neuro-rehabilitation." In Rehabilitation Robotics (ICORR), 2011 IEEE International Conference on, pp. 1-6. IEEE, 2011.
o Kahn, Leonard E., Peter S. Lum, W. Zev Rymer, and David J. Reinkensmeyer. "Robot-assisted movement training for the stroke-impaired arm: Does it matter what the robot does?." Journal of rehabilitation research and development 43, no. 5 (2006): 619.
o Colombo, R., I. Sterpi, A. Mazzone, C. Delconte, and F. Pisano. "Robot-aided neurorehabilitation in sub-acute and chronic stroke: Does spontaneous recovery have a limited impact on outcome?." NeuroRehabilitation 33, no. 4 (2013): 621-629.
o Balasubramanian, Sivakumar, Roberto Colombo, Irma Sterpi, Vittorio Sanguineti, and Etienne Burdet. "Robotic assessment of upper limb motor function after stroke." American journal of physical medicine & rehabilitation 91, no. 11 (2012): S255-S269.
o Balasubramanian, Sivakumar, Alejandro Melendez-Calderon, Agnes Roby-Brami, and Etienne Burdet. "On the analysis of movement smoothness." Journal of neuroengineering and rehabilitation 12, no. 1 (2015): 112.

Reviewer 2 ·

Basic reporting

1) The article is well-written and use technically correct text
2) Please cite also:

Wisneski KJ, Johnson MJ. Quantifying kinematics of purposeful movementsto real, imagined, or absent functional objects: Implications for modelling trajectories for robot-assisted ADL tasks. J Neuroeng Rehabil. 2007;4:7.
Panarese A. et al. Model-based_variables_for_the_kinematic_assessment_of_upper-extremity_impairments_in_post-stroke_patients. Journal of NeuroEngineering and Rehabilitation (2016) 13:81 DOI 10.1186/s12984-016-0187-9

which is relevant prior literature to be referenced.
3) The figure captions must be rewritten as they not provide enough information. Statistical results must be included in Figure 5.
4) The article is self-contained

Experimental design

The research is within aims and scope of the journal
The research question of this work is not well defined. The study seeks to find evidence that RMT for upper arm rehabilitation after stroke is beneficial to some extent. However, this was assessed in previous studies, with similar design. The authors must reformulate the abstract and introduction with a clear hypothesis within.
The reseach was conducted rigorously and to a high technical standard.
Methods are described with sufficient detail and information to replicate.

Validity of the findings

Without a clear hypothesis stated it is difficult to say whether the presented results are a 'pointless' repetition of well-known, widely accepted results.

The authors find significant improvement - on average - in the scores of clinical scales, which is to be expected because all patients are in their subacute phase. How can the authors disentangle from such result the effects of RMT?

The idea of a standard procedure based on kinematic analysis of a reaching task to compare the results of different protocols is not new (see Arch Phys Med Rehabil. 2014 Feb;95(2):338-44. doi: 10.1016/j.apmr.2013.10.006. Epub 2013 Oct 23.
Impact of time on quality of motor control of the paretic upper limb after stroke.
van Kordelaar, van Wegen, Kwakkel. Can the authors explain the added value of their research?

Data provided is robust. Conclusions are well stated.

---

## Round 0.2 · accepted · Accept

Dear Authors,

Your manuscript has been revised appropriately and is now ready for production.

Reviewer 1 ·

Basic reporting

no comment

Experimental design

no comment

Validity of the findings

no comment